# The Usefulness of the Low-FODMAP Diet with Limited Tryptophan Intake in the Treatment of Diarrhea-Predominant Irritable Bowel Syndrome

**DOI:** 10.3390/nu15081837

**Published:** 2023-04-11

**Authors:** Cezary Chojnacki, Tomasz Poplawski, Aleksandra Blonska, Paulina Konrad, Jan Chojnacki, Janusz Blasiak

**Affiliations:** 1Department of Clinical Nutrition and Gastroenterological Diagnostics, Medical University of Lodz, 90-647 Lodz, Poland; aleksandra.blonska@umed.lodz.pl (A.B.); paulina.konrad@umed.lodz.pl (P.K.); jan.chojnacki@umed.lodz.pl (J.C.); 2Department of Pharmaceutical Microbiology and Biochemistry, Medical University of Lodz, 90-236 Lodz, Poland; 3Department of Molecular Genetics, Faculty of Biology and Environmental Protection, University of Lodz, 90-236 Lodz, Poland; janusz.blasiak@biol.uni.lodz.pl

**Keywords:** irritable bowel syndrome, l-tryptophan, 5-hydroxyindoleacetic acid, kynurenine, kynurenic acid, quinolinic acid

## Abstract

(1) Background: A low-FODMAP diet is often recommended in the treatment of irritable bowel syndrome, but it does not improve abdominal symptoms in all patients, and an alternative diet is desirable. The purpose of this study was to evaluate the efficacy of a low-FODMAP diet with a concomitant reduction in tryptophan (TRP) intake in irritable bowel syndrome with diarrhea predominance (IBS-D) in relation to its metabolism via the serotonin and kynurenine pathways. (2) Methods: 40 healthy people (Group I, Controls) and 80 patients with IBS-D were included in the study. IBS-D patients were randomly divided into two groups of 40 each (Groups IIA and IIB). In Group IIA, the low-FODMAP diet was recommended, while in Group IIB, the same diet was recommended but with limited TRP intake for 8 weeks. The TRP intake was analyzed with the use of the nutritional calculator. Abdominal complaints were assessed using the Gastrointestinal Symptom Rating Scale (GSRS-IBS), and psychological status was simultaneously determined using two scales: the Hamilton Anxiety Scale (HAM-A) and the Hamilton Depression Scale (HAM-D). TRP and its metabolites: 5-hydoxyindoleacetic acid (5-HIAA), kynurenine (KYN), kynurenic acid (KYNA), and quinolinic acid (QA) were measured in urine using liquid chromatography tandem mass spectrometry (LC-MS/MS). (3) Results: The consumption of TRP per mg/kg/b.w./24 h has decreased in Group IIA from 20.9 ± 2.39 to 17.45 ± 2.41 (16.5%) and in Group IIB from 21.3 ± 2.33 to 14.32 (34.4%). Significantly greater improvement was found after nutritional treatment in patients in Group IIB as compared to Group IIA (GSRS score: 38.1% vs. 49.8%; HAM-A: 38.7% vs. 49.9%; HAM-D: 13.8% vs. 35.0%; *p* < 0.01). Reducing TRP intake showed a negative correlation with the degree of improvement in the GSRS score. (4) Conclusions: Lowering the TRP content in a low-FODMAP diet may be useful in treating IBS-D.

## 1. Introduction

Several pathogenic factors are responsible for the occurrence of irritable bowel syndrome (IBS), and food products are present among them. The human diet contains various nutrients, which may have beneficial or unfavorable effects. Numerous patients with IBS attribute their abdominal symptoms to food, and in fact, certain foods ingested can cause exaggerated gastrointestinal complaints. Patients have a tendency to avoid them and even exclude some products from their diets to relieve these symptoms. As a result, the optimal restrictive diet for IBS patients is constantly sought [1,2]. Certain diets, such as gluten-free or very low-carbohydrate or fructose-free diets, are only effective for some patients, especially those with diarrhea-predominant irritable bowel syndrome (IBS-D) [3,4,5]. Another diet without lactose was also recommended, but the results of some recent studies suggest that the symptoms of IBS-D are independent of lactose digestion [6,7]. The diet most commonly recommended for IBS-D patients in recent years is the low-fermentable oligosaccharides, disaccharides, monosaccharides, and polyols (known as the low-FODMAP) diet [8,9,10]. According to the current AGA Clinical Practice Update, the treatment of IBS-D by the low-FODMAP diet is composed of 3 distinct phases: restriction, reintroduction, and personalization [11]. In the restriction phase, the intake of FODMAP is reduced to determine whether complaints can be linked to this diet. This phase should be considered a test to determine whether IBS-D patients are sensitive to FODMAPs. Patients who respond to the restriction phase in 2–6 weeks proceed to the FODMAP reintroduction phase in the next 6–10 weeks. During this phase, FODMAP restriction is continued and simultaneously challenged with food containing a single FODMAP for 3 days, and the patient’s tolerance to diet is recorded. This information is used in the personalization phase to diversify FODMAP intake and develop an individualized diet for long-term use [12]. A low-FODMAP diet improves IBS-D symptoms in about half of patients and requires control of the composition of recommended products [13,14]. Patients without improvement in IBS-D symptoms are considered “nonresponders” and therefore require an alternative treatment [15,16,17]. We decided to modify this diet to decrease tryptophan (TRP) intake by excluding food products containing a high content of TRP. The rationale for the TRP-reduced, low-FODMAP diet was the higher serotonin levels previously observed by us and others in IBS-D patients [18,19,20,21]. In a preliminary study, we found that lowering the content of the TRP diet may reduce not only serotonin synthesis but also some neurotoxic kynurenines and improve overall symptoms in patients with IBS-D [22]. In addition, many foods in the low-FODMAP products contain a large amount of TRP, for example, some dairy products, hard cheeses, light and dark meats, and others. The purpose of the present study was to evaluate the effect of a low-FODMAP diet with a limitation of TRP intake on the clinical symptoms of patients with IBS-D in relation to its metabolism along the serotonin and kynurenine pathways.

## 2. Materials and Methods

### 2.1. Participants

120 subjects, recruited in 2017–2021, participated in this study. The control group included 40 healthy subjects without any complaints (marked as Group I, controls). The 80 patients were divided into two groups based on nutritional intervention (low-FODMAP or low-TRP and FODMAP diet). The Gastrointestinal Symptom Rating Scale (GSRS-IBS), the Hamilton Anxiety Rating Scale (HAM-A), and the Hamilton Depression Rating Scale (HAM-D) were used to analyze abdominal symptoms and the mental state of subjects as described previously [21,22,23,24]. In accordance with the standards adopted for the European population, the GSRS-IBS scale includes the following symptoms, which are scored between one and seven points: abdominal pain, pain relieved by bowel action, bloating, passing gas, constipation, diarrhea, loose stool, urgent bowel movement, incomplete intestinal emptying, fullness shortly or long after eating, and visible distension. The HAM-A and HAM-D scales distinguish mild (10–19 points), moderate (20–29 points), and severe (above 30 points) anxiety or depression.

Rome IV criteria for selecting patients were as follows: loose or watery stools occurred at least 25% of the time for six months. All patients experienced abdominal pain associated with bowel movements (the number of loose stools ranged from 3 to 8 per day); in addition, they complained of bloating. The physiological condition had a direct effect on the psychological state of the patients, in whom anxiety, lowered mood, and sleep disturbances were observed. Psychosomatic complaints persisted or recurred despite symptomatic treatment from 14 months to 6 years. Further factors for inclusion in the study group include the intensity of abdominal pain and diarrhea above 4 points on the GSRS, the score being above 24, and the results of HAM-A and HAM-D being above 11 points. Other than IBS-D diseases of the GI tract, all patients were excluded with endoscopic and histological examination of gastric, duodenal, small intestinal, and colonic mucosa. We also ruled out small intestinal bacterial overgrowth (SIBO) with the results of the lactulose hydrogen breath test (Gastrolyzer, Bedfont, Ltd., Harrietsham, UK). We excluded from the current study all patients who had a diagnosis of colitis, celiac disease, Crohn’s disease, allergy or food intolerance, parasitic and bacterial diseases, liver and renal diseases, diabetes, severe anxiety, or depression. Exclusion criteria also included the use of any drugs that could interfere with the aims of this study (including antibiotics, probiotics, and psychotropics) in the month prior to enrolment in the study.

Prior to undertaking the study, ethical clearance was obtained from all subjects. The study was conducted according to the guidelines of the Declaration of Helsinki and the Guidelines for Good Clinical Practice and approved by the Bioethics Committee of the Medical University of Lodz (RNN/176/18/KE).

### 2.2. Laboratory Tests

All subjects in the study had the routine laboratory tests performed as described previously [21,22]. Moreover, the latex agglutination photometric immunoassay was used to determine the serum concentration of C-reactive protein (CRP) using COBAS INTEGRA 800 (Roche Diagnostic, Basel, Switzerland) and an ELISA test to determine fecal calprotectin (FC) using Quantum Blue Reader (Buhlmann Diagnostics, Amherst, NH, USA). TRP and its metabolites in urine were evaluated as described previously [21,22] using liquid chromatography with tandem mass spectrometry (LC–MS/MS) in accordance with the manufacturer’s instructions (Ganzimmun Diagnostics AG, Mainz, Germany; D-ML-13147-01-01, accepted by the European Parliament—No 765/2008). The levels of the analyzed compounds were standardized at mg per gram of creatinine (mg/gCr). We also calculated the ratios of 5-HIAA and TRP as well as KYN and TRP. The 5-HIAA/TRP ratio reflected the activity of the SER pathway, whereas the KYN/TRP ratio reflected the activity of the KYN pathway.

### 2.3. Nutritional Intervention

Each participant in the study was asked to maintain a food diary, where the type and amount of food consumed daily for 14 days were recorded. A nutritional calculator from the Kcalmar.Pro-Premium app (Hermex, Lublin, Poland) was used to estimate the average daily TRP intake based on the National Institute of Public Health’s guidelines. Patients were advised to follow a balanced diet with a caloric value of 2000 kcal containing a minimum of 50 g of protein, 270 g of carbohydrates, and 70 g of fat. On the day of evaluation, everyone was given a diet with a pre-calculated TRP content. Then the patients were randomly divided into two groups of 40 each, meaning IIA and IIB. For Group IIA, the FODMAP diet was recommended for 8 weeks, after educational instruction. The patients in Group IIB were instructed to follow a low-TRP and FODMAP diet with a reduction of tryptophan intake by at least 25% compared to the initial results for 8 weeks. In the recommended diet, optimal amounts of protein, carbohydrates, and fats were maintained, and a reduction in TRP intake was achieved by excluding or drastically reducing products high in TRP such as wheat bread, sweets, hard cheeses, light and dark meats, and fish, as well as raw fruits and vegetables. The use of any drugs was forbidden. Patients from both groups received a list of all consumed products with TRP and caloric value per 100 g of product. Patients remained under dietitians’ supervision at the entire time and maintained food diaries meticulously. Caloric intake and tryptophan intake were analyzed weekly to assess adherence. After 8 weeks, follow-up medical examinations with assessment of psycho-somatic symptoms and routine laboratory tests were performed. The research was performed as an open-label clinical trial.

### 2.4. Statistical Analysis

The normality of the data distribution was checked using the Shapiro–Wilk W test. The Student’s *t*-test and the U Mann-Whitney’s test were used to compare differences between groups. Differences within groups before and after treatment were analyzed by the Wilcoxon signed-rank test. The correlation between the quantitative variables was analyzed using the Spearman rank test. Sample size was calculated at 34 cases per group with the Sample Size Calculator. Cronbach’s alpha analysis was used to calculate the internal consistency of questionnaires. The overall values of internal consistency were around 0.65, which is considered acceptable. Differences were considered significant at *p* < 0.05. All statistical analyses were performed with STATISTICA 13.3 software (TIBCO Software Inc., Palo Alto, CA, USA).

## 3. Results

Both groups did not differ in clinical parameters or daily TRP consumption (Table 1).

We did not notice significant differences in the TRP intake in both groups; however, the urinary levels of the following TRP metabolites: 5-HIAA, KYN, and QA were higher in the IBS-D group (Table 2).

All IBS-D patients in both subgroups showed similar severity of abdominal complaints as well as anxiety and depressive symptoms before nutritional intervention (Table 3).

After nutritional intervention, TRP intake was reduced from 20.8 ± 2.10 mg/kg b.w. to 17.5 ± 2.62 mg/kg b.w. (16.2%) in Group IIA and from 21.7 ± 2.16 mg/kg b.w. to 13.09 mg/kg b.w. (39.7%) in Group IIB.

In both IBS-D subgroups, a significant reduction in somatic and mental symptoms was obtained after dietary treatment (Table 4). However, these changes were more favorable in the subgroup with limited TRP intake. Calculated percentages of symptom improvement were significantly higher in the subgroup with TRP limitation and were 49.8 vs. 38.1 for GSRS, 49.9 vs. 38.7 for HAM_A, and 35 vs. 13.8 for HAM-D, respectively.

In subgroup IIB we also noticed higher number of IBS-D patients in whom the main abdominal symptoms were reduced after nutritional intervention: abdominal pain 40.5% vs. 63.2% (chi2 = 4.05, *p* = 0.044) and diarrhea—34.3% vs. 64.7% (chi2 = 6.24, *p* = 0.012) as compared to group IIA (Figure 1).

In Group IIA, no significant changes in the results of urinary excretion of TRP and its metabolites were found, while in Group IIB, 5-HIAA levels, as well as KYN and QA levels, were significantly reduced, and KYNA levels increased (Table 5).

Small percentage reducing TRP intake showed an inverse correlation with the degree of GSRS score improvement in both groups of IBS-D patients. After nutritional intervention the intensity of abdominal symptoms shows a positive dependence on the average amount of TRP intake in both sub-groups of IBS-D patients, *p* < 0.05; *p* < 0.05, respectively (Figure 2A,B).

The values of BMI and the caloric value of diet and the results of routine laboratory tests did not change significantly after nutritional intervention in both groups. The diet was well tolerated in both groups. However, the limited consumption of some products as part of the Polish diet, such as white bread, pasta, processed meat, and others, was not gladly accepted. The patients did not report any side effects. The compliance and cooperation between patients and dietitians were correct.

## 4. Discussion

The results obtained indicate that reducing the amount of TRP content in a low-FODMAP diet improves its effectiveness in the treatment of IBS-D patients. This effect is probably related to the decrease in serotonin secretion in the gastrointestinal tract. Serotonin as a neurotransmitter strongly stimulates motility and intestinal secretion, and its excess can be the cause of chronic diarrhea and abdominal pain [25,26]. It can also be the cause of mental disorders, mainly anxiety [27]. However, low levels of serotonin can result in depression, especially with a simultaneous increase in the level of kynurenine and some of its metabolites [28,29]. Several kynurenines, mainly kynurenine and quinolinic acid, can exert a neurotoxic effect and cause mental disorders [30,31,32]. According to this, depressive symptoms can be caused by a deficiency of serotonin or an excess of kynurenines. For this reason, the simultaneous testing of both TRP metabolites is justified in IBS patients who also have mood disorders [33,34]. When evaluating the effectiveness of nutritional treatment for patients, anxiety and depressive symptoms are taken into account [35,36]. Many researchers have found an increase in serum serotonin levels in inflammatory bowel disease [37,38,39], but these results do not explain the cause of comorbid mood disorders. Furthermore, no correlation was found between serotonin signaling data and somatic and psychological symptoms [40]. In recent years, the focus has been more on TRP metabolism via the kynurenine pathway, but the results obtained are also not conclusive. It was reported that KYN levels and the KYN/TRP ratio increased in IBS patients [41]. In addition, the KYN/TRP ratio was found to be positively correlated with the severity of abdominal complaints, as well as with anxiety and depressive symptoms [42]. These results have been imprecise as a consequence of the increase in activity of the kynurenine pathway. On the contrary, other researchers found the decrease in the KYN/TRP ratio in IBS-D patients and concluded that TRP metabolism along the kynurenine pathway is inhibited, and the changes are consistent with a possible increase in the serotonin pathway [43,44]. It also showed significantly higher plasma levels of 5-HT and KYNA in IBS patients compared to healthy people [45], as well as no significant changes in plasma 5-HT levels and a significantly lower 5-HIAA/TRP ratio in mixed-IBS patients [46]. In these studies, tryptophan and its metabolites were determined in plasma or serum. In our investigation, these compounds were determined in urine, which makes it difficult to compare the results. These metabolites do not accumulate in the body and come from the transformation of tryptophan both in peripheral organs and the central nervous system. The results of our study do not support the hypothesis that increased activation of the kynurenine pathway results in relative 5-HT deficiency. The levels of 5-HIAA and kynurenines were higher in patients with IBS-D compared to the control group. After reducing TRP intake, 5-HIAA and QA decreased significantly, while KYNA excretion increased. At the same time, abdominal complaints as well as anxiety and depressive symptoms decreased. Reduction in gastrointestinal symptoms was obtained in both groups, but was significantly higher in patients with monitored TRP intake. In addition, their improvement showed a positive correlation with the reduction in their intake of this amino acid. A similar direction of correlation was found for anxiety, but only in the reference group. However, no relationship was found between tryptophan intake and depressive symptoms. Therefore, the pathogenesis of depression is more complex, and the results obtained after 8 weeks of reducing tryptophan intake do not exclude it from pathogenic factors. The degree and duration of restriction intake, as well as health safety, may be important. Limiting TRP intake is not easy for patients because it requires a change in eating habits and frequent consultations with a registered dietitian nutritionist. The minimum daily intake of TRP is considered 3.5–5.0 mg per kilogram of body weight, but in the real world it is usually much higher and depends on regional dietary habits. Our patients, after a reduction in TRP intake, were advised to average 13.09 mg/kg b.w. per day, and this amount was safe and clinically useful. Abdominal ailments and symptoms of mental disorders resolved or decreased in all patients. These beneficial effects may depend on improved tryptophan metabolism. This confirms the opinion of other researchers who recognize tryptophan as ‘essential’ for the pathogenesis of IBS [47]. This opinion does not exclude the participation of other factors in the pathogenesis of this syndrome. In these patients, somatic symptoms may worsen under psychological stress [48]. On the contrary, chronic abdominal complaints raise anxiety and worsen the mood in these patients. The results of the mental health exam could also be influenced by the constant care of dietitians and doctors. No individual nutritional intervention exists for every IBS patient. However, a diet with optimized TRP intake can be useful in the treatment of IBS-D patients. The effectiveness of the low-FODMAP diet has been documented in numerous studies; however, opinions are still expressed on the need for further research on its effects.

Our study was not a randomized controlled trial and has several limitations. TRP intake data was taken from nutritional information provided by subjects, who recorded the kind and amount of food they consumed. We assumed that the information provided was honest and reliable. It was the input for the food calculator that gave data on the TRP content in the meals the subject had. In addition, mood disorders were diagnosed of the basis on the Hamilton Anxiety and Depression Scale and no other examination was done. Another weakness of the study is the non-blinded nature of the experimental design however modification of the diet and its use took place in close cooperation with the patients, therefore its blinding was impossible.

The results may also inspire further research to determine of impact the optimal intake of TRP on the somatic and mental state of the patients with different types of IBS. Furthermore, the results did not challenge the precision of the low FODMAP diet in the treatment of IBS patients, but indicate the desirability of modulating the intake of TRP. However, the results inspire further research to determine the effect of optimal TRP use on the somatic and mental state of patients with different types of IBS. 

## 5. Conclusions

Lowering the tryptophan content of a low-FODMAP diet may be useful in treating diarrhea-predominant irritable bowel syndrome.

## Figures and Tables

**Figure 1 nutrients-15-01837-f001:**
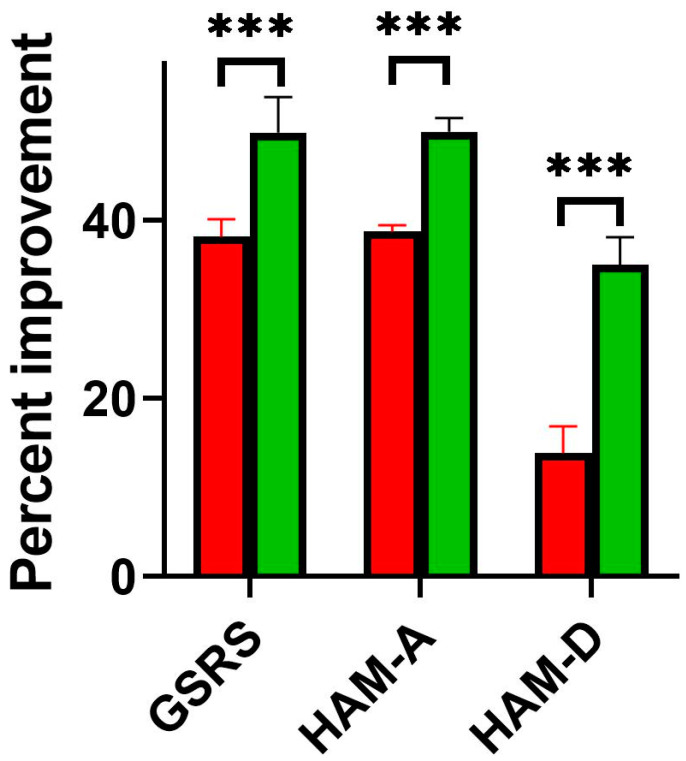
Comparison of nutritional intervention results between the two subgroups of IBS-D patients (Red bars, group IIA without and green bars, IIB with tryptophan limitation intake); GSRS—Gastrointestinal Symptom Rating Scale; HAM-A—Hamilton Anxiety Rating Scale; HAM-D—Hamilton Depression Rating Scale; Percent improvement was calculate. *** *p* < 0.001.

**Figure 2 nutrients-15-01837-f002:**
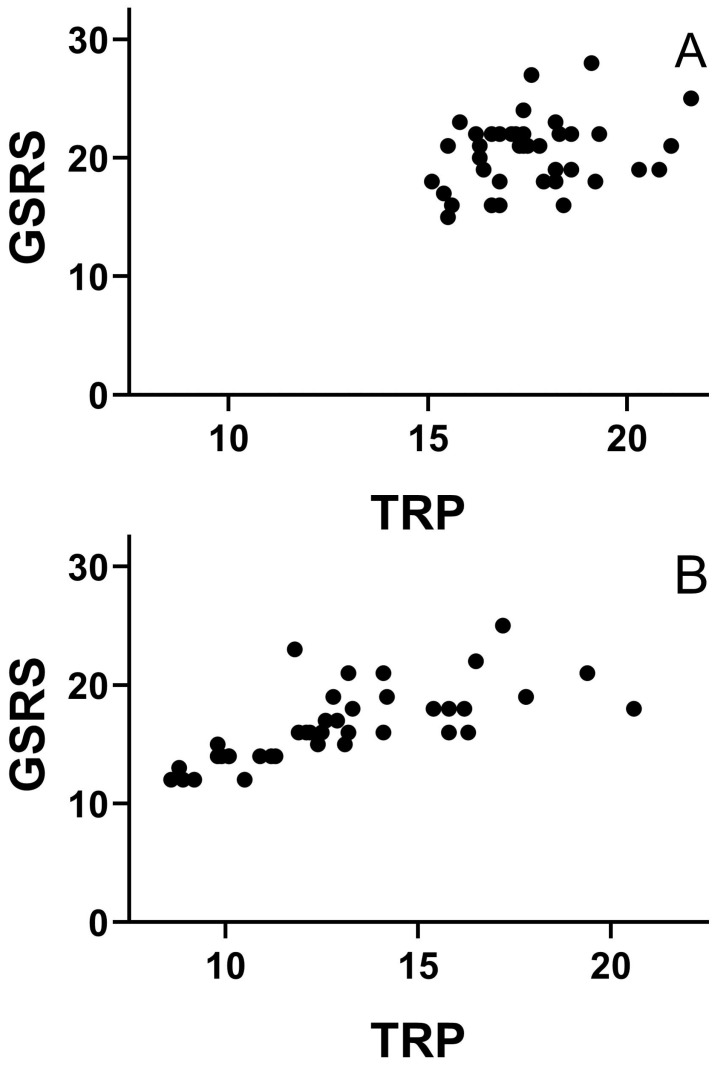
Correlation between daily TRP intake (mg/kg b.w.) and the intensity of abdominal pain (GSRS score) after nutritional treatment of IBS-D patients without (**A**) and with (**B**) an intended reduction in tryptophan intake.

**Table 1 nutrients-15-01837-t001:** General characteristics and the selected biochemical blood parameters and tryptophan intake in the control group (*n* = 40) and in patients with diarrhea-predominant irritable bowel syndrome (IBS-D, *n* = 80); average ± SD.

Feature	Group I (Control)	Group II (IBS-D)
Age (years)	46.5 ± 10.6	44.3 ± 11.4
Gender M/F	8/32	9/31
BMI (kg/m^2^)	23.5 ± 2.3	22.9 ± 1.9
GFR (mL/min)	94.8 ± 5.8	96.1 ± 4.7
ALT (µ/L)	16.8 ± 3.4	17.2 ± 4.4
AST (µ/L)	14.2 ± 2.5	15.1 ± 2.5
CRP (mg/L)	4.2 ± 1.9	5.1 ± 1.7
FC (µg/g)	31.6 ± 18.8	2.9 ± 16.9
TRP (mg/24/h)	1281 ± 146	1306 ± 129
TRP (mg/kg b.w.)	20.8 ± 2.46	21.3 ± 2.16

GFR—glomerular filtration rate, ALT—alanine aminotransferase, AST—aspartate aminotransferase, CRP—C-reactive protein, FC—fecal calprotectin; TRP—tryptophan intake; data are presented as average ± SD; differences between groups were not significant.

**Table 2 nutrients-15-01837-t002:** Urinary levels of tryptophan (TRP) and its metabolites: 5-hydroxyindoleacetic acid (5-HIAA), kynurenine (KYN), kynurenic acid (KYNA), and quinolinic acid (QA) in the control group and patients with diarrhea-predominant irritable bowel syndrome (IBS-D); all results expressed in units—mg/gCr; average ± SD.

TRP and Its Metabolites	Group I (Control)	Group II (IBS-D)	*p*-Value
TRP	11.4 ± 2.31	12.5 ± 2.21	>0.05
5-HIAA	4.92 ± 0.78	6.81 ± 0.91	<0.001
KYN	0.67 ± 0.22	0.83 ± 0.21	>0.01
KYNA	2.41 ± 0.66	2.17 ± 0.52	>0.05
QA	6.18 ± 1.02	7.25 ± 0.87	<0.01

**Table 3 nutrients-15-01837-t003:** Comparison of severity of symptoms between the two subgroups of IBS-D patients (Groups IIA and IIB); GSRS—Gastrointestinal Symptom Rating Scale; HAM-A—Hamilton Anxiety Rating Scale; HAM-D—Hamilton Depression Rating Scale; average ± SD.

Symptom Score	Group IIA	Group IIB	*p*-Value
GSRS	33.4 ± 4.11	33.3 ± 6.85	>0.05
HAM-A	20.2 ± 3.77	21.8 ± 4.38	>0.05
HAM-D	20.4 ± 3.97	18.8 ± 4.49	>0.05

**Table 4 nutrients-15-01837-t004:** Comparison of severity of symptoms between the two subgroups of IBS-D patients (Group IIA without and Group IIB with tryptophan limitation intake) before and after nutritional intervention; GSRS—Gastrointestinal Symptom Rating Scale; HAM-A—Hamilton Anxiety Rating Scale; HAM-D—Hamilton Depression Rating Scale; average ± SD.

Symptom Score	Group IIA		*p*-Value	Group IIB		*p*-Value
	Before	After		Before	After	
GSRS	33.4 ± 4.11	20.6 ± 3.10	<0.001	33.3 ± 6.85	16.4 ± 3.16	<0.001
HAM-A	20.2 ± 3.77	12.5 ± 3.59	<0.001	21.8 ± 4.38	11.1 ± 3.01	<0.001
HAM-D	20.4 ± 3.97	16.6 ± 3.55	<0.001	18.8 ± 4.49	11.3 ± 1.76	<0.001

**Table 5 nutrients-15-01837-t005:** Urinary excretion of tryptophan (TRP) and its metabolites: 5-hydroxyindoleacetic acid (5-HIAA), kynurenine (KYN), kynurenic acid (KYNA), and quinolinic acid (QA) in patients with diarrhea-predominant irritable bowel syndrome without (IIA) and with (IIB) reduction of TRP intake; before (a) and after (b) nutritional intervention; all results expressed in units—mg/gCr; average ± SD.

TRP and Its Metabolites	Group IIA			Group IIB		
	A	B	*p*	a	b	*p*
TRP	11.4 ± 2.31	10.9 ± 1.93	>0.05	12.5 ± 2.29	10.6 ± 2.06	>0.05
5-HIAA	6.31 ± 0.96	5.82 ± 1.02	>0.05	6.81 ± 0.91	3.21 ± 0.88	<0.001
KYN	0.87 ± 0.21	0.82 ± 0.19	>0.05	0.83 ± 0.21	0.80 ± 0.21	>0.05
KYNA	2.41 ± 0.66	2.74 ± 0.71	>0.05	2.17 ± 0.52	4.26 ± 0.62	<0.001
QA	7.18 ± 1.22	6.92 ± 0.98	>0.05	7.15 ± 0.87	3.90 ± 0.94	<0.001

## Data Availability

The data supporting the reported results can be provided by the corresponding authors on reasonable request.

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
