# Peer review of "The Usefulness of the Low-FODMAP Diet with Limited Tryptophan Intake in the Treatment of Diarrhea-Predominant Irritable Bowel Syndrome"

_nutrients, 2023, doi:10.3390/nu15081837_

Round 1
Reviewer 1 Report
The manuscript by Chojnacki et al. demonstrated the effect of low FODMAP Diet with limited TRP in patients with diarrhea-predominant IBS. The results of this study are interesting. There are a few issues that should be addressed prior to the publication in “nutrients”
1. Most of the comparisons in this study are between Group IIA and Group II B or between before and after the intervention. The role of control group is somewhat ambiguous. These points should be described in discussion.
2. There are no figures in this manuscript. I think it is better to make one figure of the correlation between the reduction of TRP intake and the improvement of GSRS in table 7 for readers easy to understand.
3. There are several differences in numbers between Table 3 and Table 4.
Table 3
Symptom score Group IIA Group IIB
GSRS 33.4
HAM-A 21.8
HAM-D 18.8
Table 4
Symptom score Group IIA Group IIB
GSRS 34.4
HAM-A 22.8
HAM-D 17.8
4. page 6 Table 5 → Table ‘7
Author Response
Comment (C): Most of the comparisons in this study are between Group IIA and Group II B or between before and after the intervention. The role of control group is somewhat ambiguous. These points should be described in discussion.
Answer (A): A control group was needed to rule out differences, particular in tryptophan intake and inflammatory markers. The obtained results confirm that the group of patients consists of subjects with a functional disease of GIT.
C: There are no figures in this manuscript. I think it is better to make one figure of the correlation between the reduction of TRP intake and the improvement of GSRS in table 7 for readers easy to understand.
A: The proposed figure has been inserted into the manuscript.
C: There are several differences in numbers between Table 3 and Table 4.
A: All results presented in Table 3 are correct, errors in Table 4 and text have been corrected.
Many thanks for Yours comments,
C.Chojnacki & T.Poplawski
Reviewer 2 Report
Major:
· Reducing TRP intake showed a negative correlation with the degree of improvement in the GSRS score.” This statement is inaccurate. Either say that TRP reduction showed a positive correlation with degree of improvement or that TRP intake showed a negative correlation with improvement.
· It is stated that “The currently recommended low FODMAP diet is the first-line diet”. There is concern that this is not as widely accepted in the field as this statement claims.
· “We also noticed, in group IIB, that it is more favorable, particularly with lowered levels of 5-HIAA, KYN, and QA.” The association of the measured markers with “favorability” is questionable. It is recommended that this statement be omitted in the abstract and elsewhere in the paper.
· “Conclusions: Lowering the TRP content in a low FODMAP diet is useful in treating IBS-D.” Change “is” to “may be”.
· The clinical/symptom assessment is rather unclear. This is an essential part of this study and a detailed description of exactly how this was done is needed in Methods. The intervals at which GSRS, HAM-A, and HAM-B were done needs to specified.
· Despite the various quantitative measures, this paper does not include a single plot. It would be important to show all data as scatter plots.
· Correlations also need to be shown as plots.
· Were the measurements done only at two time intervals (before and after)? Why weren’t additional time points included?
· The Discussion needs to stick with only the data. Please make this section more focused and avoid speculative statements that are not supported by the data.
· A major weakness of the study is the non-blinded nature of the experimental design.
Minor:
· Avoid the use of references in the abstract.
· The paper starts in the abstract with a rather vague and non-scientific statement: “The basis of treatment for irritable bowel syndrome is a proper diet.” What is a “proper diet”?
Author Response
Comment (C): "Reducing TRP intake showed a negative correlation with the degree of improvement in the GSRS score." This statement is inaccurate. Either say that TRP reduction showed a positive correlation with degree of improvement, or that TRP intake showed a negative correlation with improvement.
Answer (A): Agreed, this statement sounds inaccurate. In statistical terms, a small percentage reduction in TRP intake means a still high intake of this amino acid and lower degree of improvement of the ailments, which is why there is a reverse relationship. This interpretation has been supplemented in the results section. In addition, correlation between average of tryptophan intake and GSRS score in both groups has been supplemented.
C: It is stated that “The currently recommended low FODMAP diet is the first-line diet”. There is concern that this is not as widely accepted in the field as this statement claims.
A: This sentence has been removed.
C: “We also noticed, in group IIB, that it is more favorable, particularly with lowered levels of 5-HIAA, KYN, and QA.” The association of the measured markers with “favorability” is questionable. It is recommended that this statement be omitted in the abstract and elsewhere in the paper.
A: This sentence has been removed.
C: “Conclusions: Lowering the TRP content in a low FODMAP diet is useful in treating IBS-D.” Change “is” to “may be”.
A: We have done so.
C: The clinical/symptom assessment is rather unclear. This is an essential part of this study, and a detailed description of exactly how this was done is needed in Methods. The intervals at which GSRS, HAM-A, and HAM-D were done needs to specify.
A: The intervals of the GSRS, HAM-A and HAM-D have been more described.
C: Despite the various quantitative measures, this paper does not include a single plot. It would be important to show all data as scatter plots. Correlations also need to be shown as plots.
A: We have added correlations plots
C: Were the measurements done only at two time intervals (before and after)? Why weren’t additional time points included?
A: With the adopted condition, the results of nutritional intervention are evaluated after 8 Weeks.
C: The Discussion needs to stick with only the data. Please make this section more focused and avoid speculative statements that are not supported by the data.
A: Some speculative statements have been removed from the discussion.
C: A major weakness of the study is the non-blinded nature of the experimental design.
A: Disagreed, modification of the diet and its use took place in close cooperation with the patients, therefore its blinding was impossible.
C: Avoid the use of references in the abstract. The paper starts in the abstract with a rather vague and non-scientific statement: “The basis of treatment for irritable bowel syndrome is a proper diet.” What is a “proper diet”?
A: Abstract has been corrected.
Many thanks for valuable comments,
C.Chojnacki & T.Poplawski
Round 2
Reviewer 1 Report
This revised manuscript by Chojnacki et al has been extensively revised by considering the comments we previously raised. Now this manuscript appears acceptable for publication without changes any more in “Nutrients”.
Author Response
Many thanks for your insightful review
Reviewer 2 Report
A few remaining issues need to be addressed:
· Line 31-33: “Significantly greater improvement was found after nutritional treatment in patients in group IIB as compared to group IIA (GSRS score – 38.1% vs 49.8%; HAM-A–38.7% vs 49.9%; HAM-D - 13.8% vs 32 35,0%; p < 0.01).” If the authors wish to focus on these specific data as the outcome, then plots need to be provided in the body of the paper to clearly demonstrate the differences, with p values given for each individual comparison.
· P values are missing from the text in lines 216-220.
· While it is true that this specific study might not have been possible with blinding, the lack of blinding does reduce the objectivity of this study. This still needs to be addressed as in the paper as an issue.
Author Response
Comment: Line 31-33: “Significantly greater improvement was found after nutritional treatment in patients in group IIB as compared to group IIA (GSRS score – 38.1% vs 49.8%; HAM-A–38.7% vs 49.9%; HAM-D - 13.8% vs 32 35,0%; p < 0.01).” If the authors wish to focus on these specific data as the outcome, then plots need to be provided in the body of the paper to clearly demonstrate the differences, with p values given for each individual comparison.
Answer: We replaced Table 5 with plot - presented as Figure 1.
Comment: P values are missing from the text in lines 216-220.
Answer: Indeed, we supplemented the text with the missing values.
Comment: While it is true that this specific study might not have been possible with blinding, the lack of blinding does reduce the objectivity of this study. This still needs to be addressed as in the paper as an issue.
Answer: Information on this weakness was added to the manuscript in the Discussion section.